# Recent Progress in the Field of Intrinsic Self-Healing Elastomers

**DOI:** 10.3390/polym15234596

**Published:** 2023-12-01

**Authors:** Wengang Yang, Mengqi Wu, Ting Xu, Mingxiao Deng

**Affiliations:** 1Ji Hua Laboratory, Foshan 528200, China; yangwg@jihualab.ac.cn (W.Y.); wumq@jihualab.ac.cn (M.W.); 2College of Chemistry, National & Local United Engineering Laboratory for Power Batteries, Northeast Normal University, Changchun 130024, China

**Keywords:** self-healing, elastomer, intrinsic, reversible bonds, dynamic polymers, supramolecular polymers

## Abstract

Self-healing elastomers refer to a class of synthetic polymers that possess the unique ability to autonomously repair from internal and external damages. In recent years, significant progress has been made in the field of self-healing elastomers. In particular, intrinsic self-healing elastomers have garnered a great deal of attention. This mini-review outlines recent advancements in the mechanisms, preparation methods, and properties of various intrinsic self-healing elastomers based on non-covalent bond systems, reversible covalent bond systems, and multiple dynamic bond composite systems. We hope that this review will prove valuable to researchers in order to facilitate the development of novel strategies and technologies for preparing high-performance self-healing elastomers for advanced applications.

## 1. Introduction

Elastomers possess high extensibility and the capability to recover their original shape after the removal of external forces and are therefore extensively applied in various industries, including manufacturing, medical instruments, and automobile manufacturing. However, in practical applications, elastomers are often susceptible to environmental or external stresses which can lead to unexpected damage, cracks, and even macroscopic fractures, severely degrading their functionality and lifespan [1]. Self-healing elastomers, which have self-healing properties, can address these problems by fully or partially restoring in situ mechanical damages, thereby extending their service life and improving safety during use [2,3]. Consequently, self-healing elastomers have attracted tremendous attention over the past two decades as they can effectively alleviate environmental pollution, prolong the lifetime of products, and reduce costs [4,5,6,7].

Self-healing can be divided into two kinds: extrinsic self-healing and intrinsic self-healing. Extrinsic self-healing elastomers rely on pre-embedded repairing reagents (e.g., microcapsules, hollow fibers, and microvascular networks) in the matrix [8,9,10,11,12]. However, they are limited by the supply of healing agents, are incapable of multi-cycle repair and quick response, and may be affected by the outflow of healing agents [13]. On the other hand, the self-healing ability of intrinsic self-healing materials comes from the breakage and recombination of reversible chemical bonds [14]. Intrinsic self-healing elastomers do not require the addition of repair agents, thereby avoiding the above tough problems [15]. Since intrinsic self-healing elastomers not only possess superior elastomers properties but can also repair their mechanical properties themselves when physically damaged, they have become a kind of material with broad prospects [16,17].

A new generation of intrinsic self-healing elastomers has emerged based on the mechanisms of non-covalent and covalent bonds or their combinations [18]. In this mini-review article, we only focus on the mechanisms of self-healing elastomers which involve non-covalent bonds, covalent bonds, or systems which combine them (Figure 1). By summarizing a variety of successful approaches to realizing the self-healing function of elastomers over the last five years, we expect that this review will encourage more research studies in this booming field.

## 2. Non-Covalent Interactions

Non-covalent interactions are intrinsically reversible due to their lability. Introducing non-covalent cross-links into the molecular design of elastomers can result in increased mechanical strength without sacrificing extensibility, toughness, and tensile strength [19]. Non-covalent interactions that can be applied in self-healing elastomers include hydrogen bonds, ionic interactions, and metal–ligand coordination.

### 2.1. Hydrogen Bonds

Hydrogen bonds (H-bonds) have been widely used in self-healing elastomers since Cordier et al. first reported on them in 2008 [20]. The dynamic nature, tunable strength, responsiveness, and reversibility of hydrogen bonds could provide materials with significant mechanical strength and an excellent self-healing ability [21]. Low-molecular-weight polymers become robust and repairable when they are cross-linked by dense hydrogen bonds. For example, Aida et al. employed thiourea to form a less ordered hydrogen-bonded array without inducing unfavorable crystallization, and they also introduced a structural element to activate the exchange of hydrogen-bonded pairs [22]. They proposed four structural elements for the design of mechanically robust healable materials, including relatively short polymer chains for larger segmental movements, tight, cross-linked H-bonds for better mechanical properties, nonlinear (less ordered) H-bond arrays to prevent or reduce crystallization, and mechanisms for promoting the exchange of H-bonded pairs. Additionally, a biomimetic strategy was utilized to build polymer backbones with hierarchical (single, double, and quadruple) hydrogen bonding moieties (Figure 1a) [23]. The urethane, urea, and 2-ureido-4[1H]-pyrimidinone in the hierarchical hydrogen bonds endow transparent elastomers with super toughness (345 MJ m^−3^) and high tensile strength (44 MPa) after self-healing. This may be attributed to the relatively low entanglement and the dynamic exchange of H-bond pairs. Generally, dynamic dense hydrogen bonding interactions contribute to extraordinary toughness and recoverability. On the other hand, Yoshie et al. demonstrated that entropy-driven strong H-bonds enabled the creation of a vicinal diol-functionalized polymer with mechanical robustness, along with functionalities based on the dynamicity of the H-bonds [24]. Furthermore, Konkolewicz et al. even suggested that only strong and dynamic H-bonds should be chosen for enhancing materials’ performance [25].

### 2.2. Ionic Interactions

An ionic cluster is a small cluster of several to several hundred atoms or molecules with special physical and chemical properties which exist in ionomers, polyelectrolytes, and polyampholytes. Ionic clusters are associated with a comparatively long lifetime responsible for enhanced flexibility, toughness under dynamic load, and potential self-healing functions derived from sufficient molecular mobility [13]. Furthermore, ionic elastomers with dispersed ionic clusters also benefit from the variable network density of reversible aggregates. Typically, ionic interactions can be introduced into commercially available rubbers through ion–dipole interactions. A simple method that converts widely used rubbers into elastomers with extraordinary self-healing properties is replacing conventional fillers with reactive materials and constructing a reversible supramolecular hybrid network. For instance, zinc oxide (ZnO) can be blended with styrene–butadiene–styrene (SBS) or natural rubber (NR) to obtain ionic cross-links in the rubber matrix through in situ neutralization reactions [29,30]. By melt-mixing with 1-butyl imidazole, imidazolium-modified bromo-butyl rubber (bromine-modified isoprene–isobutylene copolymer, BIIR) can form a cross-linked network via ionic association [31,32]. Another strategy is to fabricate biomimetic self-healing ionic elastomers. Zhao et al. developed a type of bottle-brush elastomer with a terminal bromine atom in each side chain, resulting in the formation of a supramolecular ionic network [26]. The bottle-brush elastomer is capable of mimicking the typical features of skin by regulating the densities of branch chains and cross-linking points (Figure 1b). It exhibits a shear modulus of 46 kPa and a self-healing efficiency of 98% at room temperature. A class of self-healing strengthening elastomers (SSEs) that becomes strengthened during the healing process has also been reported [33]. This is because of the larger and denser ionic aggregates resulting from the disruption of kinetic stability under heat or an external force. These ionic self-healing elastomers hold great potential for artificial skin, wearable devices, smart soft robots, and so on [34].

### 2.3. Metal-Ligand Coordination

Metal–ligand coordination refers to the moderate bonding energy between low hydrogen bonding energy and high covalent bonding energy, providing both self-healing capacity and certain mechanical properties. The most common coordination is between the ligand and iron ions. Han and Filippidi et al. reported that reversible catechol-Fe^3+^ coordination bonds could serve as effective cross-linking points to amplify the effect of nanoscale domains and provide bonds as strong as those of covalent bonds [35,36]. Pyridine-containing ligands with Fe^3+^ can readily break and reform while the iron centers remain attached to the ligands, enabling reversible unfolding and refolding of the polymer chains [37]. Metal-coordinated polyurethane with an optimized monomer ratio and Fe^2+^ content shows a high tensile strength of 4.6 MPa (Strain ≈ 498%) and a high Young’s modulus of 3.2 MPa. Chen et al. introduced Fe^3+^-pyridine coordination bonds to rubber chains in commercially available epoxidized natural rubber (ENR) via a ring-opening reaction between epoxy groups and aminopyridine (Figure 1c) [27]. The Fe^3+^-pyridine coordination bonds can be readily broken and re-formed under moderate conditions. A sample with a molar ratio of Fe^3+^ to pyridine of 1:4 showed excellent healing efficiency with a tensile strength of 87%. Meanwhile, the Fe^3+^-pyridine coordination bonds can also act as cross-linking points, which increased the mechanical properties of the fabricated rubbers 18 times more than the original sample in terms of tensile strength. On the other hand, an Fe-triazole interaction can also achieve a healable efficiency of over 90% [38]. In addition to iron ions, Co^2+^ and Zn^2+^ have been utilized to form kinetically labile coordination bonds, endowing cross-linked polymer hydrogels, nitrile rubber (NBR), and polydimethylsiloxane (PDMS) with self-healing abilities [39,40,41,42,43]. In addition, multiple metal–ligand coordination with various metal ions can form weak or strong coordination bonds to modulate different properties of elastomers, leading to optimal self-healing efficiency and superior mechanical properties [44,45,46].

### 2.4. Other Non-Covalent Systems

Compared with the non-covalent interactions illustrated above, cyclodextrins (CDs) have emerged as promising host monomers that can construct reversible cross-linked network in elastomers via host–guest interactions [47]. For example, permethylated cyclodextrins (PM-CDAAmMe) used in the bulk polymerization of liquid acrylate monomers and guest monomers can fabricate highly flexible, self-healing, and tough elastomers [48]. In particular, the incorporation of CDs into polyurethane has demonstrated outstanding self-healing efficiency, including in waterborne polyurethane, thermoset polyurethane, and thermoplastic polyurethane, resulting in a high healing efficiency of 98.54% and nearly 100% repair after scratches [49,50,51].

Other than the host–guest interaction, shape memory has also been verified to facilitate self-healing since physical damage usually occurs during the deformation process. In blended polymer complexes, polycaprolactone (PCL) can act as a healing agent capable of diffusing and rearranging between cracks during the annealing process to heal scratches at 80 °C for 30 min as well as a semicrystalline thermoplastic (Figure 1d) [28]. It is worth mentioning that shape memory must assist self-healing with external stimuli such as thermal, magnetic, or light stimuli or a combination thereof [52]. By tailoring polymer chains, shape-memory polymers can simultaneously achieve high elasticity, excellent shape recovery, and repeatable thermal-assisted healing [53].

## 3. Covalent Bonds

Non-covalent bonds have inherent limitations due to their weak bond energy, which can easily lead to poor structural stability and mechanical properties [54,55]. In contrast, covalent bonds exhibit greater fracture tolerance compared to non-covalent bonds, thereby enhancing the mechanical strength of elastomers. Self-healing elastomers, relying on dynamic covalent bonds, have the capability to undergo dynamic bond dissociation and rearrangement in response to external stimuli [56]. Dynamic covalent bonds can be generated via several reactions, including the Diels–Alder reaction and disulfide exchange reaction.

### 3.1. Diels-Alder Reactions

Diels−Alder (DA) reactions are thermoreversible and can be applied to a wide range of elastomers. The integration of DA chemistry and rubber products has been extensively researched [57]. Most of these studies rely on the DA reactions between furan and maleimide groups [58,59]. Santana et al. employed furan moieties to graft maleated natural rubber (NR) and then cross-linked it with a bismaleimide, created a thermoreversible bridge in cross-linked NR [60] (Figure 2a). The results show that the cross-linking density and mechanical properties of the modified NR are comparable to a vulcanized NR with a low sulfur content, achieving a healing efficiency of >80% at low deformations. Rubbers like ethylene/propylene/diene rubbers (EP[D]Ms) [61], carboxylated nitrile rubber (XNBR) [62], and ENR can also be functionalized with maleimide and furan derivatives via a Diels−Alder coupling reaction [63]. These self-healing and reusable rubbers extend the product life cycle, thus reducing waste resources. As for thermoplastic self-healing polyurethane (PU) based on the DA reaction, the thermal movement effect of molecular chains accelerates the entire healing process [64]. In particular, a PU elastomer containing DA bonds and PCL soft segments can be constructed by integrating a bismaleimide (BM) with the furfuryl pendant group in PU via the DA reaction, resulting in excellent self-healing efficiency (nearly 100%) and superb mechanical strength (over 30 Mpa) [65,66]. Additionally, Zheng et al. were the first to design a PDMS elastomer network by utilizing polydimethylsiloxane and bisepoxide, both containing two DA bonds in one molecule [67]. PDMS processed through a DA modification exhibits high stretchability (over 400%), excellent self-healing properties up to 93%, and remoldability up to 95%.

### 3.2. Imine Exchange Reaction

Owing to the distinct imine exchange processes, imine bonds have been established as a form of reversible covalent bond [70]. Silicon elastomers based on PDMS are the most extensively studied imine-based elastomers [71]. Zhu et al. employed amine-functionalized PDMS as a backbone and 1,4-diformylbenzene (DFB) as a cross-linking agent, creating a reversible imine bond based on the amine groups in the PDMS while the aldehyde groups of the DFB act as self-healing points [68]. When two newly cut slices come into contact, the imine bond is rebuilt once the unbounded amino and aldehyde groups bind again, and the sample is then patched (Figure 2b). However, the healing efficiency depends on the repair time. The delicate structure design that improves chain mobility is in favor of excellent self-healing efficiency at room temperature [72]. Utilizing different aldehyde-modified tetraphenylene derivatives or boroxine derivatives as cross-linkers can even help to prepare self-healing elastomers with luminescent functionality via a dynamic imine bond [73,74]. In addition, employing a symmetric imine–diol chain extender allowed for the fabrication of a robust self-healing PU elastomer which could realize a dynamic imine exchange reaction under mild conditions [75]. The unique molecular structure led to an unexpected balance between mechanical properties and self-healing efficiency. The tensile strength was as high as 40 MPa, while the elongation reached levels of up to 880%. Its mechanical properties could nearly fully recover after 2 h of healing at 80 °C (96%). The dynamic exchange of imine bonds can also facilitate chitosan or lignin elastomers with self-healing abilities [76,77].

### 3.3. Disulfide Exchange Reaction

Disulfide bonds are easily broken, after which a new covalent bond is formed through a translocation exchange reaction to achieve self-healing in elastomers under mild conditions [78,79]. The amounts of sulfur and various disulfide/polysulfide ratios have an impact on the mechanical properties and self-healing efficiencies of the elastomer [80]. Additionally, the effects of chemical bond types and synthesis methods have also been studied [81].

A type of bio-based epoxy elastomer composed of a commercial bio-based epoxy resin (ESO) and an aromatic disulfide-containing agent (DTSA) benefits from dynamic aromatic disulfide bonds that can rapidly self-heal even at room temperature due to fast translocation exchange [82]. Another unique liquid crystal elastomer (LCE) with disulfide bonds was also previously demonstrated [83]. The polymer network rearranges under UV irradiation or upon heating, caused by a metathesis reaction of disulfide bonds, becoming reprocessable and self-healable at high temperatures.

Self-healing PUs based on disulfide bonds have attracted significant research efforts in recent years [84,85]. Liu and Zhang et al. introduced chain extenders, including 4,4′-diaminodiphenyl disulfide (DAPS) and double-hydroxyl suspended chains, into PU [69] (Figure 2c). When the usage of DAPS and suspension chains reached 30% and 50%, respectively, they induced a synergistic interaction that enabled a self-healing efficiency of up to 95%. Since the dynamic disulfide bonds of PU are embedded in the hard segments, mainly locked in the hard microphase domain, self-healable PUs with high healing efficiencies can be fabricated [86]. Taking advantage of this phase-locked bond design strategy, Dong et al. demonstrated a type of robust self-healing thermoplastic elastomer that exhibit a maximum tensile stress of 25 MPa and an elongation at break of over 1600% [87]. The scratches on the surface can recover within 60 s at 70 °C.

### 3.4. Other Covalent Systems

Other dynamic covalently cross-linked systems have also emerged [55]. For example, various boron-based bonds have been widely employed to fabricate self-healing polymeric materials [88,89,90]. Wei and Ding proposed a synergy strategy that coordinates a boronic ester and boron–nitrogen to prepare supramolecular polyurethane elastomers [91] (Figure 3a). B-N coordination accelerated the formation and dissociation of a boronic ester at room temperature, improving self-healing efficiency and also serving as a sacrificial bond to demonstrate superior notch insensitiveness and recoverability. Through a dynamic boronic ester bond, self-healing biodegradable hydrogels can be fabricated from carboxyethyl cellulose–graft–phenylboronic acid (CMC-B(OH)_2_) [92].

Otherwise, via transesterification reactions, the composite materials can be recycled and self-healed at elevated temperatures by undergoing dynamic reshuffling and rearranging the network topology [95]. The obtained sample achieved a recovery efficiency of 85% and a self-healing efficiency of 80%.

In addition, a dynamic urea-bond-based cross-linking polyurea using polydimethylsiloxane as the soft segment was developed as a self-healing elastomer [96]. The dynamic exchangeable urea bond at 60 °C enabled the dynamic exchange of the cross-linking network without altering the macroscopic shape.

## 4. Combined Systems

In general, it is hard to balance the self-healing properties and mechanical properties of materials which are essentially incompatible. At present, combining multiple interactions has become the main solution to overcome the contradiction between the mechanical properties and self-healing properties of elastomers, which has created the so-called fourth generation of self-healing materials [18].

### 4.1. Covalent Bond Based Systems

Covalent bonds possess higher energy compared to non-covalent bonds, contributing to the mechanical performance of a material. Therefore, it is crucial to enhance their reversibility to achieve high healing efficiencies by introducing non-covalent bonds or other reversible interactions [97,98].

The study of disulfide-bond-assisted H-bonding self-healing materials began systematically in 2017 [99,100]. H-bonds played a crucial role in the early stages of self-healing, while the dynamic translocation of disulfide bonds improved it by promoting polymer chain movements [101,102]. Moreover, hydrogen bonding strength can easily decline in the presence of disulfide bonds, which can result in their easier dissociation and topological network rearrangement [103,104]. Due to the synergistic interaction between stronger disulfide bonds and dynamic strong and weak H-bonds, these elastomers demonstrate high stretchability, up to 14,000%, and fast autonomous self-healing capability under universal conditions (10 min for healing at room temperature) [105].

Significantly, H-bonds are commonly used to support covalent bonds [106]. However, H-bonds and covalent bonds are intrinsically immiscible without cosolvents due to different polarities [107]. Nonetheless, it is practicable to force covalent and reversible bonds to mix at the molecular scale by cross-linking randomly branched polymers carrying motifs to create a homogenous network, resulting in tough, self-healing polymers [54]. H-bonds, urea bonds, and the products of a thiol–ene click reaction can be incorporated into a polymeric backbone with a one-pot in situ photopolymerization method, thus preparing a self-healing and robust poly(urethane-urea) elastomer [108]. Additionally, H-bonds can also cooperate with imine bonds as well [109,110].

Compared to H-bonds, ionic bonds not only have a higher bond energy but also a higher association energy due to their high propensity to segregate into ionic aggregates [111]. A dual-network structure of covalent and ionic cross-linking in ENR enables the almost unlimited mobility of rubber chains, achieving rubber-based self-healing behaviors [112]. Furthermore, imine and coordination bonds have been demonstrated to form a silicon elastomer with a dual cross-linked structure, realizing room-temperature self-healing properties with an efficiency of 94% [113].

In addition to dual-network structures, novel and facile methods incorporating multiple dynamic bonds (more than two types of bonds) have been proposed [114]. For instance, Gao et al. prepared a silicone elastomer with aminopropyl-terminated polydimethylsiloxane (A-PDMS), thioctic acid (TA), and 2,6-pyridine dialdehyde (Py) [93]. These silicone elastomers comprised disulfide bonds, hydrogen bonds, and metal–ligand bonds (Figure 3b). The hydrogen bonds formed a new weak interface while the disulfur bonds and metal–ligand bonds accelerated the formation of a strong interface. However, the lack of energy to drive hydrogen bond formation and disulfur exchange, or the lack of reactive sites, led to a lower healing efficiency at low temperatures. To achieve self-healing performance at room temperature, the synergistic effect of multiple dynamic interactions is key [115].

Furthermore, two types of dynamic reversible covalent bonds can also produce synergistic self-healing effects. The introduction of imine bonds and boroxine bonds into PDMS networks can easily fabricate a polysiloxane elastomer with a maximum healing efficiency of up to 97.8% [116]. Integrating boroxine bonds into the polymer networks can improve mechanical strength. Meanwhile, the integration of both dynamic bonds resulted in excellent self-healing properties.

### 4.2. Multiple Non-Covalent Interactions

Multiple non-covalent interactions refer to the interaction of two or more non-covalent bonds between molecules. Compared with a single non-covalent-interaction cross-linking material, multiple dynamic-interaction cross-linking polymer complexes provide a simpler method for enhancing the mechanical properties and structures of repairable elastomers.

The most common non-covalent interaction is H-bonding, which can be combined with other interactions including ionic interactions [117], coordination bonding [118,119], host–guest interactions [120], and other intermolecular forces to design self-healing elastomers with various properties.

Self-healing polymers based on ionic bonds and hydrogen bonds can achieve excellent mechanical strength while maintaining a high healing efficiency. Wang et al. utilized hydrogen bonding to prepare amide oligomers with a certain elasticity and then introduced zinc–carboxylate interactions in the oligomers [121]. The connection network was enhanced by hard ion regions and aggregation, thereby improving the rigidity and mechanical strength of the material. Additionally, this synergistic mechanism could produce highly tough and strong polyurethane/PDMS elastomers with a self-healing ability [122,123].

Regarding H-bonds and metal–ligand coordination, Qiu and Liu et al. constructed a dual physical network of H-bonds and Zn^2+^-ligand coordination which can realize fast solid–liquid separation during photocuring and 3D printing and exhibits high stretchability, a shape-programming ability, and a self-healing capacity [124]. This thermoplastic elastomer exhibited a healing strain and stress of 79% and 73%, respectively, after self-healing at 70 °C in 24 h.

Connecting H-bonds and host–guest interactions led to the construction of double networks [94]. The two networks comprised a polyacrylate matrix and poly-cyclodextrin (Poly-CD) and adamantane (Ad) groups, respectively (Figure 3c). They were miscible and interpenetrated at the molecular level due to supramolecular interactions. When the sample was stretched, the double bonds shared the load and acted as sacrificial bonds for the driving force of energy dissipation and self-healing. The sample exhibited a recovered tensile strength of up to 4.5 MPa after autonomous self-healing in ambient conditions.

In addition, H-bonds and hydrophobic associations show great potential in enhancing the mechanical performance and self-healing ability of elastomers, as indicated by Mooney–Rivlin equation calculations [125]. Furthermore, the synergistic effects of electrostatic and H-bond interactions within polymer networks could endow the complexes with a high tensile strength of 27.4 MPa and a self-healing efficiency of approximately 96% in an environment with ~90% relative humidity at room temperature [126].

## 5. Summary and Perspectives

This mini-review focuses on the latest advances in intrinsic self-healing elastomers, especially the various strategies used to construct new self-healing elastomers via different mechanisms. The self-healing elastomers prepared in this manner have a wide range of applications, including 3D printing [127], optical lenses [128], wearable electronics [129], adhesive films [130], and intelligent detection [131]. However, there are still some problems with self-healing elastomers:(1)The cost of self-healing elastomers is too high due to the high raw material cost, complex synthesis steps, and reaction conditions. In particular, the reaction intermediates typically involve highly active functional groups, such as amines, isocyanates, and free radicals, which will undoubtedly limit their shelf life and prohibit their applications.(2)So far, the repair conditions of self-healing materials are too harsh. Generally, achieving a self-healing function at room temperature is difficult, and it requires heat at a certain temperature or electromagnetic radiation to achieve self-healing. Moreover, flow and closure are the foundations that make healing possible. Thus, it is vital to put fractured structures back together, which is not currently possible without manual intervention.(3)The repair speeds of self-healing elastomers have not yet met the needs of practical applications.(4)With increases in use time and repair time, the physical properties of self-healing materials will weaken. Meanwhile, their healing speed and efficiency will be greatly reduced due to the water molecules or dust particles that may occupy the positions where reversible bonds form and break between polymer chains.(5)It is a challenge to ally robust mechanical performance with virtuous self-healing abilities in elastomers.

In view of the above limitations, future works on self-healing elastomers should focus on the following aspects:(1)Introducing self-healing properties into commercially available polymer materials without changing the original properties of the polymer material, such as PDMS, rubbers, and PUs.(2)Optimizing the position, concentration, and structural arrangement of the dynamic chemical bonds in the polymer network to realize the transition from elastomer to liquid after dynamic bond breakage so that the flow of the polymer segment can also completely fill the damaged area.(3)Improving the mobility of molecular chains and the dynamic exchange rate of dynamic bonds to shorten the healing time.(4)Integrating different chemical groups into polymer networks with fine control of the proportions and positions responsible for self-healing and other functions during synthesis or through post-modification.(5)Increasing cross-linking density and endowing the dynamic properties of cross-linking points, which may improve the mechanical properties of self-healing materials, or alternatively constructing a dual network structure composed of both non-covalent and covalent bonds into the material. Dynamic covalent bonds cause a significant improvement in mechanical properties, while non-covalent interactions, as the primary sacrificial bond, provide substantial improvements in healing efficiency. In addition, it is advisable to choose reversible covalent bonds with higher bond energies combined with auxiliary supramolecular interactions, controlling the structure of the soft segment and the soft/hard segment ratio or constructing a microphase separation system and introducing an enhanced phase. All these methods aim to achieve self-healing elastomers with both high mechanical strength and superior healing efficiency.(6)Specifically, it is beneficial to establish mechanisms by mimicking organisms found in plants and human skin when designing self-healing elastomers.(7)Exploring multiple combinations in search of further positive effects.(8)Using fillers or other additives as carriers for additional repair mechanisms may also be a promising option.

In summary, ideal self-healing elastomers should possess a quick healing time, high efficiency, robust mechanical properties, and autonomous healing without external energy and be fully restored (including mechanical properties and other functions) after repair. Additionally, developing multifunctional self-healing elastomers will be a future research hotspot. We believe that the present serious limitations will be resolved in the near future, and that diverse high-performance self-healing elastomers with various properties have excellent prospects.

## Data Availability

Not applicable.

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
