# Peer review of "Recent Progress in the Field of Intrinsic Self-Healing Elastomers"

_polymers, 2023, doi:10.3390/polym15234596_

Round 1

Reviewer 1 Report

Comments and Suggestions for Authors

The manuscript provides a comprehensive review of recent progress on intrinsic self-healing polymers and is well-written. I recommend publishing it as is.

Reviewer 2 Report

Comments and Suggestions for Authors

The field of self-healing polymers is vast. I don't see this paper cover an area in self-healing polymers that significantly adds to the field.

Comments on the Quality of English Language

Please carefully check for errors and typos. There are too many of them, such as missing spaces between sentences, etc.

Reviewer 3 Report

Comments and Suggestions for Authors

Dear Authors,

thank you very much for interesting manuscript. I find it as very interesting and very suitable for academic area. Therefore, I agree with acceptation of it in the current form form content viewpoint.

On the other hand, I see space from improvement in its presentation, namely in consistency of terminology. Sometimes you used Non-covalent, sometimes Noncovalent... etc. Please check it very carrefully in the whole body of manuscript.

Best wishes

Reviewer 4 Report

Comments and Suggestions for Authors

I suggest to accept the paper as it is. Authors  provided mini review of the progres in self healing elastomers, and this paper could be great base for further research, because it comprised of a lot referencies.

Reviewer 5 Report

Comments and Suggestions for Authors

This is a review paper. However, the authors appear to have only limited knowledge of this topic. As an example, the authors do not have a clear idea what “self-healing” is about, and the authors are not aware that there are some other mechanisms to achieve “intrinsic self-healing” as well. There are some extensive review papers in the literature about this topic and it is hard to see any new contributions in this paper. Thus, this paper should be rejected.

Here are some issues need to clarify.

-       to autonomously repair or recover from physical damage, without the need for external intervention

Well, apparently most polymetric materials mentioned in this paper cannot achieve self-healing without the need of external intervention.

-       Under the current background of promoting environmental protection, the self-healing materials can effectively reduce waste, prolong the lifetime of products and reduce costs[5]. However, elastomers are often susceptible to environmental or external stresses, which can lead to unexpected damage, cracks, and even macroscopic fracture, ultimately resulting in material failure[6].

It is very hard to follow the logic in above paragraph. Language editing of the whole paper is required.

-       can address these problems by fully or partially restoring in situ mechanical damages

English editing is required for the whole paper.

-       In the first two paragraphs and abstract, self-healing is defined again and again in different ways. The structure of this paper needs to be revised for a better flow and consistency.

-       Our review is based on the mechanism of the bonds and interactions which involved: non-covalent, covalent or combined systems between them(Scheme 1)

In the literature, there are a couple of different mechanisms for so called “intrinsic self-healing” in this paper. As mentioned above, what reviewed in this paper is only limited to some of them. Thus, the title of this paper is misleading.

Comments on the Quality of English Language

Need extensive language and technical editing.

Round 2

Reviewer 2 Report

Comments and Suggestions for Authors

The fonts in all the figures are too small. For example, in Figure 1c and d. The word 'healed' in Figure 1c is also covering the schematic.

Comments on the Quality of English Language

Please check again for typos. There are too many of them and I just listed a few here. For example, in line 85, it should be 'This may be attributed to...'. In line 90, it should be 'et al.' . In line 95, instead of 'Fe3+-pyridine', it should be 'Fe3+-pyridine'.

Reviewer 5 Report

Comments and Suggestions for Authors

Excellent elasticity (which is mostly missing in the previous studies) and good heat assisted healing have been reported in the shape memory hybrids reported in C C Wang et al 2012 Smart Mater. Struct. 21 115010

This is similar to SEBS/PCL material, but developed based on the concept of shape memory hybrid.

This type of “intrinsic self-healing” elastomer is missing in this review.

Comments on the Quality of English Language

NA
